SOFTWARE

# AIEdit: Alignment-free genome assembly polisher trained on spaced seed match patterns

Parham Kazemi[1,2], Ivana Sánchez Olivares[1,2], René L. Warren[1], Lauren Coombe[1], Inanc Birol[1,3]*

**1** BC Cancer Research Institute, Vancouver, Canada, **2** Bioinformatics Graduate Program, Faculty of Science, University of British Columbia, Vancouver, Canada, **3** Department of Medical Genetics, University of British Columbia, Vancouver, Canada

* ibirol@bccrc.ca, inanc.birol@ubc.ca

## Abstract

Polishing, the process of correcting base-level errors in genome assemblies, is a critical step for ensuring accuracy in downstream analyses, such as variant calling, gene annotation, and clinical genomics applications. While recent advances in long-read sequencing technologies have helped improve assembly contiguity and genome completeness, maintaining high base-level accuracy in those genomes remains challenging due to the still appreciable errors associated with certain long-read sequencing technologies. Existing polishing approaches face notable trade-offs: alignment-based methods achieve high accuracy but incur long run times, alignment-free k-mer-based tools are scalable but struggle in regions with dense errors, and machine learning-based polishers often only perform well on specific platforms and require read-to-assembly alignments. We present AIEdit, a machine learning-based polisher designed to operate alignment-free, generalizing across sequencing platforms while remaining computationally efficient. We developed AIEdit by combining spaced seed matching with a neural network trained to detect and correct dense error patterns in an alignment-free manner. We benchmarked the method on simulated and experimental DNA sequencing data. On simulated human long-read assemblies with high error rates, AIEdit reduced error rates by 58% compared to ntEdit's 21%, completing in 2.7 hours using 230 GB of memory, faster than POLCA and Medaka (multi-day run times) and using 3×less memory than JASPER (689 GB). On experimental Oxford Nanopore Technologies (ONT) data from the NA24385 human genome, AIEdit increased the Merqury quality score (QV) from 28.7 to 32.9 in 9.5 hours, achieving comparable accuracy to Medaka (QV 32.7) in a fraction of the time (1.5+days) and outperforming k-mer-based tools ntEdit (QV 31.0) and JASPER (QV 31.7). Overall, AIEdit enables scalable and accurate genome polishing across diverse datasets.

**Data availability statement:** AIEdit is freely available at github.com/BirolLab/AIEdit.

**Funding:** This study is supported by the Canadian Institutes of Health Research (CIHR) [PJT-183608, I.B.]. The funder had no role in study design, data collection and analysis, decision to publish, or preparation of the manuscript.

**Competing interests:** The authors have declared that they have no competing interests.

## 1 Introduction

Modern genomics relies heavily on the availability of high-quality genome assemblies [1–5]. Recent advances in sequencing technologies and computational tools have substantially improved assembly contiguity, enabling telomere-to-telomere genome reconstruction [6,7]. However, base-level errors remain common, primarily due to the inherent noise in sequencing instruments and inaccuracies in basecalling and assembly. These errors appear as mismatches and insertions or deletions (indels), usually smaller than 5 bp, and can affect the accuracy of downstream analyses. The issue is more pronounced in long-read platforms, such as those from Oxford Nanopore Technologies PLC (ONT, Oxford, UK) and Pacific Biosciences of California, Inc. (PacBio, Menlo Park, CA, USA), where reads are sequenced with higher error rates than short-read technologies [8,9]. As a result, identifying and correcting base-level errors, commonly referred to as polishing, has become an important step in genome assembly pipelines.

Many widely used polishing tools, such as Pilon [10], Racon [11], POLCA [12], and NextPolish [13], correct errors by aligning reads to the draft assembly. While these methods often achieve high polishing accuracy, the computationally expensive process of generating alignments becomes a bottleneck in genome assembly pipelines. To address this issue, alignment-free polishers have emerged as a scalable alternative. These polishers commonly rely on the frequency of short subsequences of length $k$ (k-mers) in raw sequencing reads. Typically, high-confidence k-mers that accurately represent the underlying genome appear frequently in the sequencing reads, whereas erroneous k-mers occur at lower frequencies. ntEdit [14] and JASPER [15] are two methods which use k-mer frequency profiles to identify weakly supported sequences in an assembly and replace them with high-confidence ones. Since gathering k-mer frequencies is often faster than aligning reads to the assembly, these methods scale well with larger genomes. However, k-mer-based polishers struggle in regions with dense error patterns, where multiple errors within a single k-mer break the expectation of observing exact matches and eliminate the signal needed for correction. This happens frequently in long-read data, limiting the effectiveness of k-mer-based methods for polishing long-read assemblies.

To address the sensitivity of exact k-mer matching, alternative representations such as spaced seeds [16] and strobemers [17] have been developed. Spaced seeds tolerate base mismatches by marking specific positions in a k-mer as "do not care", allowing the match to depend only on other positions. Strobemers extend this idea by linking short non-adjacent subsequences, determined randomly or deterministically, to improve robustness to indels. These structures have been shown to outperform traditional k-mers in various tasks such as *de novo* assembly, read alignment, indexing, and sequence mapping [18,19]. Consequently, spaced seeds and strobemers offer a promising direction for polishing assemblies from long reads.

Recent studies have also explored the application of machine learning (ML) in assembly polishing. These methods typically learn systematic error patterns in different sequencing platforms. For instance, PEPPER [20] and Medaka (github. com/nanoporetech/medaka) use deep neural networks to identify and correct errors

in assemblies generated from ONT data. These models are especially effective at platform-specific errors, such as homopolymer compressions. These ML-based polishers rely on read alignments during both training and inference, inheriting many of the same scalability issues as other alignment-based methods. The potential of alignment-free ML-based approaches for polishing genome assemblies remains largely unexplored.

In this work, we have developed AIEdit, a novel, alignment-free, ML-based polishing pipeline. By using spaced seed match/mismatch patterns as input rather than raw sequencing content or read alignments, AIEdit's model learns generalizable error signatures that transfer datasets and sequencing platforms without retraining. We benchmarked AIEdit against widely used alignment-based, k-mer-based, and ML-based polishers, to demonstrate its accuracy and computational efficiency.

## 2 Design and implementation

AIEdit's pipeline consists of five main stages (Fig 1). It begins by extracting high-confidence k-mers and spaced seeds from the input sequencing reads. These k-mers are used to identify low-confidence regions in the draft assembly, which are then analyzed by a neural network that predicts the most likely error pattern based on spaced seed membership. Erroneous bases are replaced with supported alternatives, followed by a final polishing step using ntEdit.

### 2.1 Error region location

To identify high-confidence k-mers, AIEdit first generates a k-mer count histogram and selects a count threshold, typically the first local minimum (S1a Fig), that separates low-abundance (assumed erroneous) k-mers from genomic ones. It then constructs two sets with elements exceeding this threshold: one for high-confidence k-mers and one for high-confidence spaced seeds. These sets are implemented as Bloom filters [21], probabilistic data structures that support fast membership queries using a fixed-size bit array and multiple hash functions, with a controlled false positive rate and no false negatives. S1b Fig illustrates that these false positives have minimal impact on the overall polishing accuracy. AIEdit scans the draft assembly for stretches of consecutive k-mers absent from the Bloom filter. These regions are pushed into a shared queue by the error region detector. Multiple consumer threads then pop from the queue in parallel, using the error pattern detector to infer base-level error patterns within each region, marking them for correction.

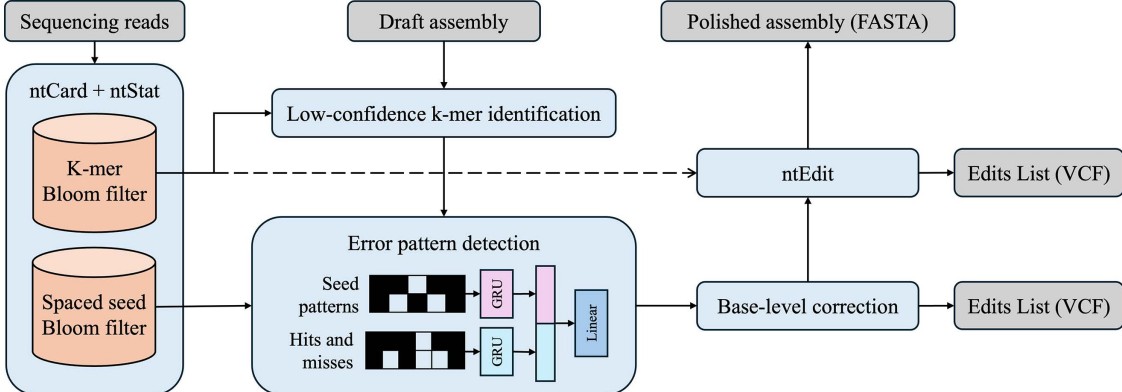

**Fig 1. Overview of the AIEdit polishing pipeline.** In the pre-processing stages, ntCard and ntStat construct Bloom filters (orange cylinders) containing high-confidence k-mers and spaced seeds from sequencing reads (upper left grey box). These Bloom filters are used to identify unsupported k-mers in the assembly and form the input to the error pattern model. Black (white) cells in the inputs to the error pattern model represent spaced seed "care" ("do not care") positions, and patterns and hits are passed to the pink and cyan GRUs, respectively. The final hidden states of these two GRUs are concatenated and passed to the linear layer (dark blue) for predicting the error pattern, which guides base-level corrections. Finally, a single round of ntEdit is applied as a post-processing step to correct potentially missed errors, reusing the k-mer Bloom filter. Input and output files are shown in grey boxes.

## 2.2 Error pattern detection

The error pattern detector is a neural network consisting of two input Gated Recurrent Units (GRU) [22] followed by an output linear layer (Fig 1). The first GRU receives the structure of the spaced seeds as a 2-dimensional tensor, $S$, where $S_{i,j} = 1$ if the $i$th base of the $j$th seed is a care position, and 0 otherwise. This representation allows the model to generalize across different seed structures and k-mer sizes. The second GRU receives a 2-dimensional tensor $X$, which encodes local context. To help the model capture homopolymer-associated error patterns, the first column of this tensor, $X_{i,0}$, contains a binary flag indicating whether the $i$th base in the region is identical to the one immediately preceding it. Subsequent rows of the tensor encode spaced seed membership information for different gap sizes to allow the model to detect indels. Specifically, for each gap size $g \in [0, w]$, where $w$ is a hyperparameter representing the maximum detectable indel length, $X_{i,wj+g} = 1$ if spaced seed $j$ in position $i$ is still present in the set of high-confidence seeds after introducing a gap of $g$ bases at the start of the region, and 0 otherwise. These gaps cause matches when they overlap with the "do not care" positions in a spaced seed. After processing the inputs, the GRU hidden states are concatenated and passed through the output layer to predict edit operations, including substitutions and indels.

The arrangement of "care" positions directly influences the error patterns that can be detected by the model. Therefore, we implemented a genetic algorithm for generating seed patterns that maximize the support of diverse error signatures. The algorithm begins with a randomly initialized set of seed patterns and iteratively optimizes them over a fixed number of generations. The objective function includes the number of distinct error patterns detectable by the seed set while penalizing low-weight seeds by incorporating the number of "care" positions. The final optimized seed set is used during preprocessing and model training.

## 2.3 Error correction

In this stage, AIEdit applies edits that maximize the presence of high-confidence k-mers in each region. It begins by deleting bases marked for removal and inserting placeholder gaps at positions flagged as insertions. Then, it permutes the gaps and positions marked as substitutions with alternative bases. From the top $T$ error patterns ranked by model confidence, we select and apply edits that improve the high-confidence k-mer ratio to at least $p$. The highest-ranking acceptable set of edits is applied to the assembly, and other edits are reported as potentially heterozygous variants. If no improvement is observed, the region remains unedited and is recorded as filtered entries in the VCF output. During this process, AIEdit also identifies assembly gaps (represented as runs of Ns in the draft assembly) as potential error regions and attempts to resolve them if the gap size is within the model's supported mismatch signature or indel size.

The parameters $T$ and $p$ can be specified by the user and are empirically set to 3 and 0.5 by default, respectively. While the default of $T = 3$ is sufficient for most assemblies, increasing it to $T = 10$ can improve results in noisy datasets by increasing the number of attempted edit patterns at the cost of a slight increase in runtime (Section 3.3). Parameter $p$ defines the stringency of the correction, where a higher value (e.g., $p = 0.8$) ensures that edits are only applied if they are supported by 80% of the k-mers in that region. This higher stringency is particularly effective for reducing false-positive corrections in low-coverage or highly heterozygous assemblies.

## 2.4 ntEdit integration

To capture any residual errors not detected by the neural network, AIEdit performs a final post-processing step using ntEdit. In this stage, the partially polished assembly and the k-mer Bloom filter are passed to ntEdit, a lightweight polisher optimized for correcting sparse errors. Our analyses show that this step effectively corrects indels missed in earlier stages with minimal added computational cost compared to rerunning the entire pipeline (S3 Fig).

## 2.5 Implementation

AIEdit's preprocessing stage uses ntCard [23] to generate the k-mer count histogram. It then uses ntStat's [24] histogram analysis module to determine the count threshold at the intersection of the erroneous and genomic k-mer distributions. Next, ntStat's filtering module constructs the sets of high-confidence k-mers and spaced seeds. We use the Bloom filter implementations from the btllib library [25], which are compatible with ntEdit v2.0 and later, allowing the k-mer set to be reused for the post-processing stage. The error region detector, error pattern model, and base correction components are implemented in C++ and exposed to Python using pybind11 [26]. The model is trained using a Python script implemented with PyTorch [27] and compiled into a TorchScript module, which is loaded for polishing and used in C++ through libtorch.

## 2.6 Data and experiment setup

To train the error pattern detector, we created five sets of spaced seeds containing three seeds, for each k-mer size between 25 and 60 base pairs (bp). For validation, we generated two additional seed sets for k-mer sizes divisible by 5 between 25 and 50 bp. We found that using three seeds per set provided a balance between the number of detectable error patterns and the computational cost of the preprocessing stage. Finally, we simulated all possible error patterns with a window size of $w = 5$ bp for the ground truth training labels and applied these error patterns to randomly generated substrings from the human genome [5]. The inputs to the model were formed as expected under ideal conditions, assuming no Bloom filter false positives. The model was trained for 100 epochs using the AdamW optimizer [28] with a learning rate of 0.001. The hidden state size of the GRUs was set to 32. We recorded the training history of the model (S2 Fig) and saved the model with the highest validation accuracy for use in polishing. We evaluated several model configurations (1–4 layers, 16–128 hidden units) and found that increasing capacity beyond our defaults yielded negligible improvements in accuracy. Consequently, we prioritized a lightweight architecture to balance performance with computational scalability. Alternative pretrained models are provided in our repository for users seeking different configurations.

We evaluated AIEdit's polishing accuracy, speed, and memory footprint by simulating 250 bp paired-end short reads with 0.1% error rates from the *Caenorhabditis elegans* N2 strain [29], *Drosophila melanogaster* [30], and *Homo sapiens* T2T-v2.0 [5] reference genomes using pIRS [31]. The *C. elegans* dataset was generated at 50-fold coverage, and the *D. melanogaster* and *H. sapiens* datasets were generated at 30-fold coverage. To simulate draft assemblies, we used pIRS to introduce errors in the reference genomes with 0.1% mismatch and 0.01% indel rates.

In addition, we generated long-read datasets with a 0.6% approximate error rate from the *D. melanogaster* and *H. sapiens* genomes using NanoSim (v3.2.2, Kit v14 dorado model) [32], simulating 80-fold and 60-fold coverage, respectively. The simulated long reads were then assembled using GoldRush (v1.2.2) [33]. We applied polishing to both the contigs and scaffolds of the simulated long-read assemblies to test the tools' effectiveness in the different stages of the assembly pipeline.

To evaluate AIEdit on experimental sequencing data, we gathered publicly available ONT and PacBio datasets and assembled the reads using GoldRush (S1 Table). These datasets include ONT reads of *D. melanogaster* (accession: SRR22822929), sequenced at 154-fold coverage and basecalled with Guppy v6.4.2 with an estimated 2.9% error rate; the ONT dataset and assembly of the NA24385 (HG002) human cell line, as published in the GoldRush study, sequenced at 67-fold coverage using R9.4.1 pore chemistry and basecalled with Guppy v5.0.6 with a 4.0% error rate; and PacBio Circular Consensus Sequencing data from the NA24385 human cell line with a 0.2% error rate [34].

We compared AIEdit's polishing performance against ntEdit and JASPER as k-mer-based polishers originally developed for short-read and long-read data, respectively. We also included POLCA and NextPolish as alignment-based methods and Medaka as an ML-based polisher. All tools were executed with their default parameters on a DELL server with 144 Intel Xeon Gold 6254 CPUs at 3.10 GHz with 3.1 TB of RAM. We limited the maximum CPUs available for each experiment's process to 72. Guided by the results shown in S1 Fig, we set the default k-mer size in AIEdit to 35 and used

this value for all experiments. ntEdit was executed using the same k-mer size as AIEdit, as we passed the k-mer Bloom filter directly from the AIEdit framework (**Fig 1**). JASPER was run with its default k-mer size of 37. Polishing accuracy, including the number of mismatches and indels in the assembly per 100 kbp, was evaluated using QUAST (v5.3.0) [35]. For the experimental dataset, we used Merqury [36] in addition to QUAST for reference-free evaluation. We also recorded the total run time and peak memory usage of each tool across its entire pipeline. For AIEdit, this includes the time and memory used by ntCard, ntStat, the polishing stages, and the final ntEdit round. Command-line arguments and tool versions are provided in S2 Table.

## 3 Results

The full benchmarking results used for illustrating Fig 2, including mismatch and indel error rates, run time, and memory usage for simulated datasets, as well as mismatch and indel rates with Merqury QV scores for experimental datasets, are presented in S3-S5 Tables.

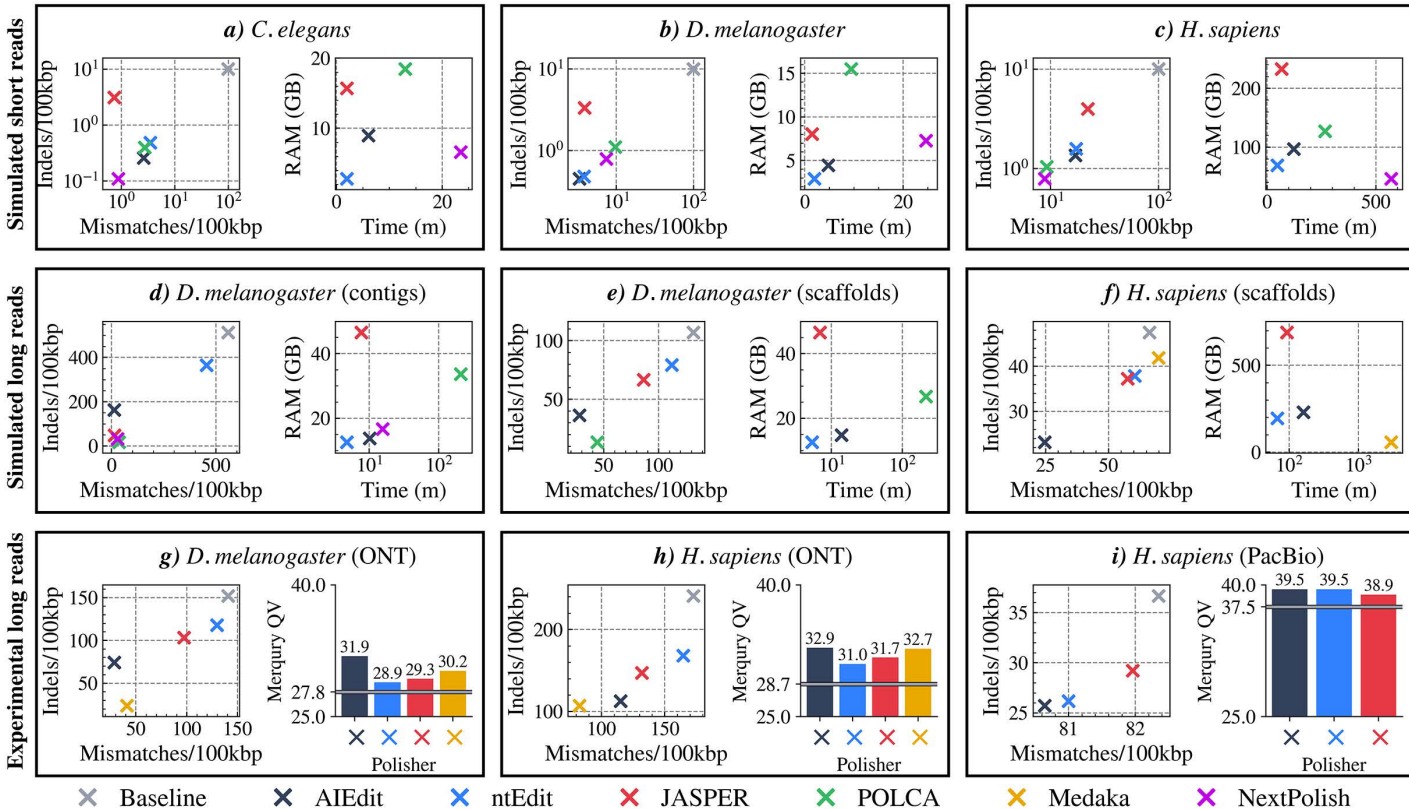

**Fig 2. Polishing accuracy and computational performance of AIEdit compared to other tools.** Polishing accuracy is reported as the average number of mismatches and indels per 100kbp. The number of mismatches and indels in the unpolished assemblies are shown as grey circles, and their QV scores are shown as grey horizontal lines. For simulated short-read datasets (a, b, and **c)**, AIEdit (dark cross), ntEdit alone (blue cross), and POLCA (green cross) achieved similar error correction rates. Their results are shown in logarithmic scale to highlight minor differences. The run times for the simulated long-read experiments (d, e, and f) are also in log-scale due to POLCA and Medaka's (orange cross, when available) high run times compared to other tools. All other plots are in linear scale. "Baseline" (grey cross) refers to the unpolished assembly. Run time and peak memory usage includes the computational resources allocated for each tool's entire pipeline reported by/usr/bin/time -pv. For experimental long read datasets (g, h, and **i)**, k-mer QV scores of the polished assembly are calculated by Merqury.

## 3.1 Simulated short reads

For the simulated *C. elegans* assembly from short-read data (Fig 2a), AIEdit performed comparably to ntEdit and POLCA, with all three tools reducing error rates by 96–98%. While JASPER achieved the highest mismatch correction rate at 99.3%, its indel correction rate was 28.5% lower than AIEdit. NextPolish corrected 98.9% of the indels, the highest correction rate in this experiment. In the simulated *D. melanogaster* assembly (Fig 2b), AIEdit corrected 96.6% of mismatches and 95.5% of indels, less than a 1% difference to ntEdit. AIEdit also outperformed JASPER in indel correction by 27.3% and showed a 6.6% higher mismatch correction rate than POLCA. For the *H. sapiens* assembly (Fig 2c), POLCA and NextPolish had the highest overall accuracies, where POLCA reduced the mismatch rate by 90.8% and NextPolish corrected 92.1% of the indels. AIEdit and ntEdit again performed similarly, and AIEdit outperformed JASPER in this experiment by 5.1% and 26.2% in mismatch and indel reduction, respectively.

## 3.2 Simulated long reads

The contigs from the *D. melanogaster* assembly from simulated long reads (Fig 2d) initially contained 558.99 mismatches and 513.14 indels per 100 kbp. AIEdit reduced these errors by 97.7% and 68.2%, slightly outperforming JASPER in mismatch correction by 0.2% and POLCA by 4.1%. While POLCA achieved a higher indel correction rate, it required 3.48 hours to complete, over $20\times$ longer than AIEdit which finished in 10.11 minutes. Moreover, AIEdit used 13.8 GB of memory, 59.1% less than POLCA and 70.3% less than JASPER. Despite setting NextPolish's mapping mode to minimap2 to reduce run time, it still took $1.5\times$ longer than AIEdit to complete. At the scaffold level (Fig 2e), AIEdit reduced mismatches and indels by 78.8% and 65.9%, respectively, outperforming the other k-mer-based polishers. Specifically, its mismatch and indel correction rates were 69.3% and 40.0% higher than those of ntEdit, and 44.3% and 22.4% higher than JASPER. While POLCA corrected 21.5% more indels than AIEdit, it corrected 12.3% fewer mismatches. In terms of computational performance, AIEdit required $15\times$ less time and $3.1\times$ less memory than POLCA and used $1.8\times$ less RAM than JASPER.

AIEdit corrected the most errors from the scaffold-level *H. sapiens* assembly (Fig 2f), achieving mismatch and indel reductions of 62.9% and 51.3%, respectively. ntEdit and JASPER were less accurate, reducing mismatches by 8.7% and 13.0%, and indels by 20.3% and 21.6%. In terms of run time, AIEdit's pipeline completed in 161.2 minutes, $2.4\times$ longer than ntEdit and $1.7\times$ longer than JASPER. All k-mer-based tools were substantially faster than POLCA, which did not complete after three days. Regarding memory usage, AIEdit used 229GB of RAM, 17.4% more than ntEdit but $3\times$ less than JASPER, which consumed 689GB.

NextPolish was excluded from scaffold-level benchmarks due to its documented limitation at processing assemblies containing gaps. While NextPolish offers higher accuracy than POLCA with faster runtimes on contig-level assemblies, we opted not to fragment our scaffolds to accommodate its requirements, ensuring all tools were evaluated on identical, structurally intact assemblies.

To further characterize AIEdit's performance, we evaluated the precision ($P$) and recall ($R$) of its corrections across various genomic contexts in the scaffold-level *H. sapiens* assembly (S4 Fig). AIEdit demonstrated consistent robustness in challenging regions, including low GC content (≤35%) and homopolymers (≥ 5 bp). Notably, in repetitive regions such as centromeres and segmental duplications, defined using annotations from the UCSC Genome Browser and the T2T-CHM13 project, AIEdit maintained high precision and recall ($P=0.79$, $R=0.75$ and $P=0.70$, $R=0.59$), effectively identifying and fixing dense error patterns in complex regions.

We also verified that AIEdit's performance is not specific to the GoldRush assembly profile and evaluated its accuracy on assemblies generated by Flye (S5 Fig). AIEdit achieved comparable error reduction on these datasets, demonstrating its ability to generalize across different assembly algorithms.

## 3.3 Experimental long reads

As shown in Fig 2g, AIEdit polished the draft *D. melanogaster* by reducing mismatches by 79.7%, which was 71.2% more than ntEdit and 48.8% more than JASPER. Similarly, AIEdit corrected 51.3% of indels, outperforming ntEdit by 28.8% and JASPER by 19.3%. To assess base-level accuracy, we built a 21-mer database from high-quality Illumina reads (accession: SRR11460799) using Meryl and evaluated the assemblies using Merqury. AIEdit achieved a QV of 31.9, an improvement over ntEdit (28.9) and JASPER (29.3). Once again, we terminated POLCA after three days of run time without completion.

The unpolished *H. sapiens* assembly from ONT sequencing data contained 172.66 mismatches and 241.01 indels per 100kbp, using the maternal chromosomes of the HG002 reference genome [37] as ground truth. In this experiment, increasing AIEdit's stringency parameter to $p = 0.8$ and the number of top-ranking patterns evaluated to $T = 10$, improved polishing accuracy by reducing false-positive corrections. As shown in Fig 2h, AIEdit reduced mismatches and indels by 33.4% and 53.2%, respectively, outperforming ntEdit (28.7% and 23.0%, respectively) and JASPER (9.8% and 14.2%, respectively). Although Medaka achieved the highest polishing accuracy based on QUAST, reducing mismatches by 52.5% and indels by 55.6%, it required over 1.5 days to complete, while AIEdit finished in 9.5 hours. AIEdit also improved the Merqury QV from 28.7 to 32.9, higher than ntEdit (31.0), JASPER (31.7), and Medaka (32.7). The higher QV for AIEdit, despite its slightly higher error rates, likely reflects limitations in QUAST's single-haplotype alignment, whereas Merqury captures base-level accuracy across both haplotypes via k-mer matching.

Assembling the experimental PacBio dataset of the NA24385 genome resulted in a baseline draft assembly containing 82.35 mismatches and 36.67 indels per 100kbp. Similar to the low-error simulated short-read experiments, all tools showed comparable polishing accuracy. AIEdit reduced mismatch and indel rates by 2.1% and 29.8%, respectively, with most of its advantage over JASPER observed in indel correction, outperforming it by 9.5%. Merqury analysis showed a baseline QV of 37.5, with both AIEdit and ntEdit improving it to 39.5, while JASPER reached a slightly lower QV of 38.9.

## 3.4 Discussion

We developed AIEdit to address three limitations in existing assembly polishers. First, alignment-based polishers such as POLCA deliver high accuracy but require computationally expensive read-to-assembly alignments. Second, alignment-free k-mer-based polishers like ntEdit and JASPER scale well but lose error-detection signal when multiple errors occur within the same k-mer, a frequent scenario in long-read assemblies. Third, current ML-based polishers, such as Medaka, can effectively capture systematic sequencing errors but require platform-specific training and depend on alignments, inheriting the same scalability issues as traditional alignment-based methods.

AIEdit demonstrates how the scalability of alignment-free methods can be augmented with machine learning. By gathering well-supported spaced seeds in a Bloom filter, AIEdit avoids the computational overhead of read alignment. Spaced seeds also address the limitation of traditional k-mer-based approaches by allowing matches in the presence of multiple errors within a single k-mer, preserving signal in error-dense patterns.

Detecting base-level error patterns from spaced seeds is analytically intractable due to the complex relationship between seed design, k-mer size, genomic repeats, and Bloom filter false positives. Therefore, we framed the task as a sequence approximation problem and designed an error pattern model based on the theoretical foundation that recurrent neural networks are universal sequence-to-sequence approximators [38,39]. The model is trained to detect a wide range of error signatures across different k-mer sizes and seed configurations independent of the actual sequence content, making it generalizable to all assemblies. As a result, unlike existing methods that require specific training for different sequencing platforms and chemistries, AIEdit is platform-agnostic.

The effectiveness of the error pattern model depends on the design of the spaced seeds, since error detection becomes possible when a mismatch pattern aligns with the "care" positions of at least one seed in the set. For instance,

the mismatch pattern M*MM*, where 'M' indicates a mismatch and '*' denotes a match, can belong to the Bloom filter if one of the spaced seeds in the set has the substring 01001, with 1 representing "care" positions. To maximize the support of possible error patterns, AIEdit uses a spaced seed generator based on a genetic algorithm. This approach allows efficient exploration of the large search space of possible seed patterns, selecting those that maximize the number of supported error patterns while maintaining seed weight for specificity.

In datasets with sparse base-level errors, such as short-read assemblies, we found that lightweight tools like ntEdit provide similar accuracy with faster run times. To take advantage of this, we included ntEdit as a final post-processing step in the AIEdit pipeline. This allows AIEdit to efficiently correct remaining errors in a second round of polishing with marginal overhead. Among the tools we benchmarked, ntEdit was consistently fast, memory-efficient, and accurate for polishing short-read assemblies. Moreover, its Bloom filter implementations are fully compatible with AIEdit, enabling seamless integration. We investigated whether running the full AIEdit pipeline for multiple rounds would yield further improvements but found that assembly quality remains largely unchanged when using the same k-mer size. While a multi-round strategy using varying k-mer sizes is theoretically possible to target different error densities, the vast search space and the overhead of multiple k-mer size runs would result in substantially longer run times.

Our stratified analysis (S4 Fig) reveals that while AIEdit is highly effective across most of the genome, it can occasionally introduce incorrect indels in regions of high complexity, such as large segmental duplications or long centromeric repeats. This is a known limitation of k-mer-based methods that utilise a small context window (typically $2 \times k$), as the repetitive nature of these regions can mask the local signal required for unambiguous error pattern prediction. However, overall, the gain in base-level accuracy and the significant reduction in computational overhead compared to alignment-based polishers make AIEdit a highly practical solution for large-scale genomics.

Our results show that within the current landscape, AIEdit is faster than alignment-based and ML polishers, yet more accurate than conventional k-mer-based approaches for high-error long-read data. Given these advantages, we anticipate AIEdit's primary utility to be in polishing assemblies from ONT data, where dense error patterns are more common than other sequencing technologies. Overall, AIEdit provides a novel solution for alignment-free ML-based genome polishing, enabling accurate, large-scale downstream genomics analyses.

## 4 Availability and future directions

High assembly accuracy is essential for downstream applications, yet base-level errors remain common, especially in long-read sequencing data. Existing polishing tools face fundamental trade-offs: alignment-based methods achieve high accuracy at the cost of computational efficiency, k-mer-based approaches scale well but fail when multiple errors cluster within single k-mers, and machine learning methods require platform-specific training and read alignments. AIEdit addresses these limitations by combining spaced seed matching with recurrent neural network-based error detection in an alignment-free framework. By operating on abstract spaced seed patterns rather than sequencing-specific signals, AIEdit generalizes across platforms without retraining. Our benchmarking demonstrates that AIEdit achieves competitive or superior accuracy on high-error long read assemblies while completing substantially faster than alignment-based and ML methods. AIEdit provides a practical solution for scalable, accurate genome polishing across diverse sequencing technologies and is freely available at https://github.com/BirolLab/AIEdit.

## Supporting information

**S1 Fig. Sensitivity analysis of AIEdit's k-mer size and Bloom filter false-positive rate parameters.** (a) AIEdit's polishing accuracy across count thresholds in k-mer histograms, with colors representing different k-mer sizes. Short-read data and assemblies were simulated from the *D. melanogaster* reference genome using pIRS, and long-read datasets were generated using NanoSim and assembled with GoldRush. (b) Impact of Bloom filter false-positive rates (FPR) on error

rates in the simulated long-read assembly. The plot demonstrates the model's robustness to membership noise, showing that even a 100-fold increase in FPR results in a negligible increase in residual mismatches and indels (<3 per 100kbp).
(PNG)

**S2 Fig. Training history of the error pattern detection model. a) Training and validation loss across 100 epochs. b) Validation accuracy across training epochs.** Training accuracy was not calculated to reduce training time.
(PNG)

**S3 Fig. Impact and overhead of running ntEdit in AIEdit's pipeline on simulated long-read data from the *D. melanogaster* genome. a) Wall clock run time of each stage of the pipeline. b) Polishing accuracy of each configuration: AIEdit, AIEdit without ntEdit (--no-ntedit flag), AIEdit twice with the --no-ntedit flag, and ntEdit alone.**
(PNG)

**S4 Fig. Stratified precision and recall of AIEdit polishing for the assembly of long reads simulated from the T2T-v2.0 human reference genome.** These results were gathered by aligning the draft and polished assemblies to the reference genome using minimap2, followed by variant calling with bcftools and intersection analysis using bcftools isec. Confusion matrices show the number of fixed, missed, and introduced errors across all errors (a-e), mismatches (f-j), and indels (k-o). Analysis is stratified by whole assembly, low GC content, homopolymers, centromeres/satellites, and segmental duplications. Genomic coordinates for repeats and segmental duplications were obtained from the UCSC Genome Browser and the marbl/CHM13 repository. The intensity of the blue colors are proportionate to the value in each cell normalized by the total number of edits in each confusion matrix (fixed + missed + introduced).
(PNG)

**S5 Fig. AIEdit's polishing performance on Flye assemblies of the experimental *D. melanogaster* data in different stages compared to other tools.** (a) Base-level accuracy as measured by Merqury for unpolished and polished contigs and scaffolds. AIEdit demonstrates a consistent improvement in QV scores, comparable to or exceeding established polishers. (b) Number of mismatches and indels per 100kbp. AIEdit effectively reduces both error types across all assembly stages, confirming its generalizability to different assembly algorithms and error profiles.
(PNG)

**S1 Table. Contiguity and correctness of the assemblies generated from experimental long-read data using GoldRush, reported by QUAST.**
(XLSX)

**S2 Table. Tool versions and command-line arguments used for experiments.** Paths to the draft assembly, sequencing reads, and Meryl database are replaced with ASSEMBLY, READS, and MERYL_DB, respectively. The estimated genome size is passed to GoldRush with G = SIZE.
(XLSX)

**S3 Table. Polishing accuracy reported by QUAST and performance of the tools on the simulated short-read assemblies.**
(XLSX)

**S4 Table. Polishing accuracy and performance of the tools on the assemblies from simulated long reads.**
(XLSX)

**S5 Table. Polishing accuracy of the tools on the assemblies from experimental long reads.**
(XLSX)

## Author contributions

**Conceptualization:** Parham Kazemi, Inanc Birol.

**Data curation:** Parham Kazemi, René L. Warren, Lauren Coombe.

**Funding acquisition:** Inanc Birol.

**Methodology:** Parham Kazemi, René L. Warren, Inanc Birol.

**Resources:** René L. Warren.

**Software:** Parham Kazemi, Lauren Coombe.

**Supervision:** Inanc Birol.

**Validation:** Ivana Sánchez Olivares.

**Visualization:** Ivana Sánchez Olivares.

**Writing – original draft:** Parham Kazemi.

**Writing – review & editing:** Ivana Sánchez Olivares, René L. Warren, Lauren Coombe, Inanc Birol.

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
