## [Decision Letter · Decision Letter 0]

13 Jan 2026

PCOMPBIOL-D-25-02236

AIEdit: alignment-free genome assembly polisher trained on spaced seed match patterns

PLOS Computational Biology

Dear Dr. Birol,

Thank you for submitting your manuscript to PLOS Computational Biology. After careful consideration, we feel that it has merit but does not fully meet PLOS Computational Biology's publication criteria as it currently stands. Therefore, we invite you to submit a revised version of the manuscript that addresses the points raised during the review process.

We look forward to receiving your revised manuscript.

Kind regards,

Isidore Rigoutsos

Academic Editor

PLOS Computational Biology

Ilya Ioshikhes

Section Editor

PLOS Computational Biology

**Journal Requirements:**

3) Your manuscript is missing the following sections: Design and Implementation, and Availability and Future Directions. Please ensure that your article adheres to the standard Software article layout and order of Abstract, Introduction, Design and Implementation, Results, and Availability and Future Directions. For details on what each section should contain, see our Software article guidelines:

https://journals.plos.org/ploscompbiol/s/submission-guidelines#loc-software-submissions

5) We notice that your supplementary Figures, and Tables are included in the manuscript file. Please remove them and upload them with the file type 'Supporting Information'. Please ensure that each Supporting Information file has a legend listed in the manuscript after the references list.

**Reviewers' comments:**

Reviewer's Responses to Questions

**Comments to the Authors:**

Reviewer #1: This manuscript presents AIEdit, an alignment free assembly polishing pipeline that combines spaced seed membership signals from Bloom filters with a GRU based model to predict local edit patterns, followed by a final ntEdit round. The idea is promising, especially for long read assemblies where dense errors reduce the effectiveness of exact k-mer methods.

Major revision requests

1. Evaluate further

For simulated data, report precision and recall of corrections separated into substitutions, insertions, deletions, and homopolymer contexts.

2. Importance of AI components

Show AIEdit without the ML model (heuristic substitute) to show what the actual contribution is.

3. Quantify robustness to Bloom filter false positives and parameter sensitivity

Training assumes ideal membership without Bloom filter false positives, but inference does not. Sweep Bloom filter false positive rate and show impact. Now it reads like dataset specific tuning.

4. Stratify performance in difficult genomic contexts

Show error reduction stratified by repeats, low complexity, GC extremes, and homopolymer length.

5. Generalize beyond one assembler

Evaluate on at least one additional assembler output to show the method is not overly coupled to a specific assembly error profile. Clarify behavior on contigs vs scaffolds and gap handling. Also, what about metagenomics?

Minor points

Clarify the operational definition of “alignment free” (especially relative to evaluation), and present reference-based vs reference free metrics with consistent interpretation.

Overall, the method is interesting, but it needs a stronger experimental backbone and clearer attribution of improvements before the claims feel conclusive.

Reviewer #2: The paper by Kazemi et al. describes AIEdit, a new alignment-free polisher for genome assemblies that uses ML. Polishing assemblies is an important problem in genome assembly pipelines to ensure accuracy in downstream genomics analyses, such as variant calling or genome annotation.

AIEdit detects regions that need be corrected using low-abundance k-mers (contiguous and spaced) implemented as Bloom filters. Then, it uses a neural network (two GRUs followed by a linear layer) to detect errors. AIEdit uses a spaced seed generator

based on a genetic algorithm to maximize the support of possible error patterns.

The manuscript is well-written and the content is interesting from a computational viewpoint.

Major issues:

(A) Parameters.

(1) The choice of k appears to be good for AIEdit according to Suppl Figure 1, but what about the other tools? How was k chosen for the other tools?

(2) The choice of the hyper-parameters for the neural network should to be discussed, e.g., layers, # of units, etc

(3) The choice of the parameter T and p needs to be discussed. The manuscript says "The parameters T and p can be specified by the user and are set to 3 and 0.5 by default." How are the results depends on the choice of these parameters? Under which circumstances users should change these parameters? Please discuss

(B) Experiment results.

(1) In the manuscript AIEdit is compared against ntEdit, JASPER, POLCA and Medaka. NextPolish and Polypolish should be included for completeness. These two latter tools are considered good polishing tools (see, e.g., https://doi.org/10.1186/s12864-024-10582-x)

(2) Most polishing tools are known to introduce errors, not just correct them. What type of errors does AIEdit tend to introduce?

(2) The authors of some genome polishing tool recommend to use them repeatedly (2-3 rounds). Would running AIEdit more than once bring any benefit?

(3) Can the author comment on the idea of using a pipeline of two different polishing tools to improve the quality of the assemblies even more?

**Have the authors made all data and (if applicable) computational code underlying the findings in their manuscript fully available?**

Reviewer #1: Yes

Reviewer #2: Yes

PLOS authors have the option to publish the peer review history of their article (what does this mean?). If published, this will include your full peer review and any attached files.

Reviewer #1: No

Reviewer #2: No

**Figure resubmission:**
---

## [Decision Letter · Decision Letter 1]

17 Apr 2026

Dear Dr. Birol,

We are pleased to inform you that your manuscript 'AIEdit: alignment-free genome assembly polisher trained on spaced seed match patterns' has been provisionally accepted for publication in PLOS Computational Biology.

Best regards,

Isidore Rigoutsos

Academic Editor

PLOS Computational Biology

Ilya Ioshikhes

Section Editor

PLOS Computational Biology

Reviewer's Responses to Questions

**Comments to the Authors:**

Reviewer #1: Authors have addresses all of the comments - I think the manuscript can be published as is

Reviewer #2: The authors have satisfactorily addressed all my comments. Thanks for your work.

**Have the authors made all data and (if applicable) computational code underlying the findings in their manuscript fully available?**

Reviewer #1: Yes

Reviewer #2: Yes

PLOS authors have the option to publish the peer review history of their article (what does this mean?). If published, this will include your full peer review and any attached files.

Reviewer #1: No

Reviewer #2: No

---

## [Editor Report · Acceptance letter]

PCOMPBIOL-D-25-02236R1

AIEdit: alignment-free genome assembly polisher trained on spaced seed match patterns

Dear Dr Birol,

I am pleased to inform you that your manuscript has been formally accepted for publication in PLOS Computational Biology. Your manuscript is now with our production department and you will be notified of the publication date in due course.

With kind regards,

Anita Estes
